# Prevalence and correlates of Human Papillomavirus infection in females from Southern Province, Zambia: A cross-sectional study

Lweendo Muchaili[1]*, Precious Simushi[1], Bislom C. Mweene[1], Tuku Mwakyoma[1], Sepiso K. Masenga[1,2], Benson M. Hamooya[1,3,4]

1 HAND Research Group, School of Medicine and Health Sciences, Mulungushi University, Livingstone Campus, Zambia, 2 Department of Medicine, Vanderbilt University Medical Center, Nashville, Tennessee, United States of America, 3 School of Public Health, University of Alabama at Birmingham, Birmingham, Alabama, United States of America, 4 Centre for Infectious Disease Research in Zambia, Lusaka, Zambia

* muchailiL2@gmail.com

**Data Availability Statement:** All relevant data are within the manuscript and its Supporting Information files.

## Abstract

### Background

Human papillomavirus (HPV) infection is strongly associated with cervical cancer with almost all cases being associated with the infection. Cervical cancer is the leading cause of cancer death among women in Zambia and the fourth leading cause of cancer death in women worldwide. However, there is limited data on the burden and associated factors of HPV in sub-Saharan Africa. This study therefore aimed to determine the prevalence and correlates of HPV infection in the Southern province of Zambia.

### Methods

This was a cross-sectional study conducted at Livingstone University Teaching Hospital (LUTH) among 4,612 women from different districts of the southern province being screened for HPV infection between September 2021 and August 2022. Demographic and clinical data were collected from an existing laboratory programmatic database. Multivariable logistic regression was used to estimate the factors associated with HPV infection.

### Results

The study participants had a median age of 39 years [interquartile range (IQR) 30, 47]. The prevalence of HPV infection was 35.56% (95%CI). At multivariable analysis, the factors associated with a positive HPV result were younger age (adjusted odds ratio (AOR) 0.98; 95% confidence interval (CI) 0.98–0.99; p. value 0.001), having provider collected sample (AOR 2.15; 95%CI 1.66–2.79; p. value <0.001) and living with HIV (AOR 1.77; 95%CI 1.22–2.55; p. value <0.002).

**Funding:** The author(s) received no specific funding for this work.

**Competing interests:** The authors have declared that no competing interests exist.

## Conclusion

The prevalence of HPV in women in the southern province of Zambia is high, and likely influenced by age and HIV status. Additionally, the outcome of the HPV test is affected by the sample collection method. Therefore, there is a necessity to enhance HPV and cervical cancer screening, especially among people with HIV.

## Introduction

Human papillomavirus (HPV) infection is strongly associated with cervical cancer and remains the most common pathogen associated with cancer in females [1]. Some studies have shown that by the age of 50, about 80% of the world's population is exposed to HPV infection thus making HPV the most common sexually transmitted infection [2,3]. The prevalence of cervical-vaginal HPV worldwide is 9.9% with most of the burden being in low- and middle-income countries (LMICs) [1]. The pooled HPV prevalence in Sub-Saharan Africa (SSA) is estimated to be 32.3% in women [4]. Sub-Saharan Africa region has one of the highest HPV burdens in the world and consequently has higher cervical cancer cases [4,5].

Cervical cancer is the fourth leading cause of cancer death in women worldwide. It is estimated that 570,000 females were diagnosed with cervical cancer and 311,000 globally died of cervical cancer in 2018 [6]. Approximately 84% of all cervical cancers and 88% of deaths due to cervical cancer occur in LMICs [7]. SSA reports around 75,000 new cervical cancer cases and 50,000 deaths from the disease every year [8]. According to the World Health Organization (WHO) projections, cervical cancer deaths will reach 443,000 by the year 2030 and 90% will be from the sub-Saharan region [9].

Cervical cancer has been one of the most prevalent types of cancer for than a decade, and a leading cause of cancer death in women in Zambia affecting about 97.1 per 100,000 women [10,11]. An estimated 3000 women in Zambia are diagnosed with cervical cancer yearly [12]. In 2018, Zambia had the third highest cervical cancer incident rate worldwide at 66.4 per 100,000 with a total of 1,839 women dying of cervical cancer [13]. A study done by Hayumbu et al. in 2021 at Livingstone University Teaching Hospital (LUTH), found the prevalence of cervical cancer among women attending the cancer clinic to be 6% with an additional 19% of the women having precancerous cervical lesions [14].

Several studies have demonstrated that HPV infection is associated with early initiation of sexual activity, multiple sexual partners, having other sexually transmitted infections, use of oral contraceptives, and smoking [15,16]. Low socioeconomic status, education level, and living with HIV have also been shown to be strongly associated with HPV infection [17–20]. However, there is a paucity of specific data on the burden and drivers of HPV infection in SSA. This study aimed to determine the prevalence and correlates of HPV infection among women from Southern province using programmatic data in order to generate hypotheses for follow-up interventional studies.

## Methods

### Study design and setting

This was a retrospective cross-sectional study conducted at Livingstone University Teaching Hospital (LUTH) from 5th May 2023 to 24th May 2023. LUTH is the regional referral hospital for the Southern province of Zambia. Southern province is the third largest province in

Zambia with a population of over 2.3 million people and covering 85,283 square kilometers. The study utilized programmatic data for the period of September 2021 to August 2022, during which the LUTH laboratory was the sole facility performing HPV Polymerase Chain Reaction (PCR) testing in the province. All health facilities in the region collected and sent cervical swabs to the LUTH laboratory for HPV PCR testing.

### Eligibility criteria

We included all women who had been screened for HPV infection in the period September 2021 to August 2022. We excluded all women who had missing HPV PCR results from the study.

### Study population

We counted both the electronic entries on the Laboratory Information System (LIS) and the physical forms stored in the laboratory to ascertain the total number of women tested; about 5,100 women were tested for HPV infection using PCR in this period. After excluding those with missing outcome variable, we remained with a final sample size of 4, 612 women.

### Study variables

The primary outcome variable was the HPV infection. Explanatory variables included HIV status (Living with HIV or Living without HIV), sample type (Provider-collected or Self-collected), visit type, and entry point [Visual Inspection with Acetic Acid (VIA) routine check-up, antiretroviral therapy (ART) clinic or other hospital sections].

### Visit type

Women were categorized based on their clinic visit number. Those attending the clinic for cervical cancer screening for the first time made the 1st category, and those visiting the clinic the second, third, and fourth time made the 2nd, 3rd, and 4th categories respectively.

### Data sources/ measurement

A researcher-designed quantitative tool was used to collect data from the programmatic data provided by the referring health facilities. Data on the following variables were collected; HIV status, sample type, visit type, entry point and HPV result. All study participants were examined using the Aptima® HPV kit on the Hologic Panther platform. This assay is for the qualitative detection of E6/E7 mRNA transcripts for 14 high-risk HPV types (16, 18, 31, 33, 35, 39, 45, 51, 52, 56, 58, 59, 66, and 68). All data was collected and analyzed anonymously.

### Data analysis

Collected data were entered, assorted, coded, and then analyzed using Stata version 15. Descriptive statistics (i.e. proportions, frequency, medians, and interquartile range (IQR)) were used to understand the distribution of study variables. The Shapiro-Wilk test was used to test for the normality of continuous variable (age). Wilcoxon rank-sum test was used to compare the medians for non-normally continuous data (age) between the two outcome groups (HPV positive and HPV negative). A Chi-square test was used to ascertain a relationship between two categorical variables. The logistic regression model (bivariate and multivariable) was used to determine factors associated with HPV infection. A p-value of less than 0.05 was considered to be statistically significant.

## Results

### General characteristics of participants

The study comprised 4,612 female study participants with a median (IQR) age of 39 years (30, 47). The majority (n = 2,734, 59.28%) were women living with HIV (WLWH) of which 1,960 (42.50%) were referred to the cancer clinic by the ART clinic. Most participants (n = 3,421, 74.18%) were visiting the cancer clinic for the first time and had their samples collected by a healthcare provider (n = 2,599, 56.35%). HPV PCR was positive for 1,640 (35.56%) of the participants, see Table 1.

### Relationship between HPV infection and clinical and demographic characteristics

Women with HPV PCR positive results were significantly younger than those without (median age 37 vs 39 years, p = 0.002). HPV infection was significantly higher in women who had their samples collected by a healthcare provider compared to those who had self-collected samples (36.98% vs 21.4%, p = <0.001), and in WLWHIV compared to women living without HIV (36.06% vs 25.45%, p = <0.001), Table 2.

### Univariable and multivariable analysis

Univariate analysis shows that an increase in age by a year had a 1% reduction in the odds of having HPV infection (OR 0.99; 95%CI 0.98–0.99; p = 0.014). Women who were referred to the cancer clinic by other hospital sections had 38% reduced odds of being infected with HPV compared to those who went for routine VIA (OR 0.62; 95%CI 0.40–0.95; p = 0.031). Women who had provider-collected samples were 2.15 times more likely to have a positive HPV result

**Table 1. Demographic and clinical characteristics of participants.**

| Variable | Median (IQR) | Frequency | Percent (%) | Missing data, n (%) |
|---|---|---|---|---|
| **Age (years)** | 39 (30, 47) | | | 384 (8) |
| **Visit type**<br>1st | | 3,421 | 97.83 | 1,115 (24) |
| 2nd | | 5 | 0.14 | |
| 3rd | | 66 | 1.89 | |
| 4th | | 5 | 0.14 | |
| **Entry point**<br>VIA routine checkup | | 991 | 32.28 | 1542 (33) |
| ART clinic | | 1,960 | 63.84 | |
| Other | | 119 | 3.88 | |
| **Sample type**<br>Self-collected | | 444 | 14.59 | 1569 (34) |
| Provider collected | | 2,599 | 85.41 | |
| **Living with HIV?**<br>No | | 279 | 9.26 | 1599 (35) |
| Yes | | 2,734 | 90.74 | |
| **HPV result**<br>Negative | | 2,972 | 64.44 | 0 |
| Positive | | 1,640 | 35.56 | |

IQR: Interquartile range.

Note: Some variables had fewer than 4612 entries due to missing values on the medical record.

**Table 2. Relationship between HPV infection with clinical and demographic characteristics.**

| Variable | Number | HPV result | | p-values |
|---|---|---|---|---|
| | | Positive, 1,640 (35.56%) | Negative, 2,972 (64.44%) | |
| **Age (years)), median (IQR)** | 4,228 | 37 (29, 47) | 39 (31, 47) | **0.002[w]** |
| **Visit type** | 3,497 | | | 0.309 |
| 1st | | 1,177 (34.41%) | 2,244 (65.59%) | |
| 2nd | | 0 (0) | 5 (100%) | |
| 3rd | | 19 (28.79%) | 47 (71.21%) | |
| 4th | | 2(40%) | 3(60%) | |
| **Entry point** | 3,070 | | | 0.093 |
| VIA routine check-up | | 358 (36.13%) | 633 (63.87%) | |
| ART clinic | | 684 (34.9%) | 1,276 (65.1%) | |
| Other | | 31 (26.05%) | 88 (73.95%) | |
| **Sample type** | 3,043 | | | **<0.001** |
| Self-collected | | 95 (21.40%) | 349 (78.60%) | |
| Provider collected | | 961 (36.98%) | 1,638 (63.02%) | |
| **Living with HIV** | 3,013 | | | |
| No | | 71(25.45%) | 208(74.55%) | **<0.001** |
| Yes | | 986 (36.06%) | 1,748(63.94%) | **<0.001** |

Abbreviations: IQR: Interquartile range, w: Wilcoxon rank-sum test, VIA: Visual inspection with acetic acid, ART: Antiretroviral therapy, HPV: Human papillomavirus, bold p-values show variables with p<0.05.

compared to those with self-collected samples (OR 2.15; 95%CI 1.69–2.74; p = <0.001). Women who were living with HIV had a 65% increased chance of having HPV infection compared to women without HIV (OR 1.65; 95%CI 1.24–2.18; p = <0.001).

At multivariable analysis, an increase in age by a year had a 2% reduction in the chance of having HPV infection (AOR 0.98; 95%CI 0.98–0.99; p = 0.001). Having a provider collected sample increased the odds of having a positive HPV diagnosis by 2.15 times when compared to self-collected samples (AOR 2.15; 95%CI 1.66–2.79; p<0.001). Participants with HIVWL-WHIV had 77% increased odds of having HPV infection when compared to ones without HIV (AOR 1.77; 95%CI 1.22–2.55; p = 0.002), see Table 3.

## Discussion

Our study aimed to investigate the prevalence and correlates of HPV infection among female study participants visiting cancer clinics in different health facilities in the southern province. The overall prevalence of HPV infection was high among our study participants (35.56%) and it was found to be more prevalent among women with HIV. This is consistent with a study conducted in Southern Africa by Ramogola-Masire et al. [21]. However, our findings are lower than some previous studies conducted in Zambia. A study conducted by Ng'andwe et al. with a sample size of 70 women found an overall HPV prevalence of 65%, 80% in WLWH, and 55% in seronegative women [22]. One reason that could explain the difference is that Ng'andwe et al. study screened for 17 high-risk HPV (hr-HPV) strains whereas our study focused on 14. Furthermore, Ng'andwe et al. used a smaller sample size compared to our study. Another study by Sahasrabuddhe et al. among WLWHIV in Zambia found hr-HPV prevalence to be 90.3% [23]. Like the study conducted by Ng'andwe et al, Sahasrabuddhe et al. included more (18) hr-HPV types in their study. Equally, the Sahasrabuddhe et al. study had a

**Table 3. Univariable and multivariable analysis of factors associated with HPV infection.**

| Variable | Crude analysis OR | 95%CI | p-value | Adjusted analysis OR | 95%CI | p-value |
|---|---|---|---|---|---|---|
| **Age (years), median (IQR)** | 0.99 | 0.98–0.99 | **0.014** | 0.98 | 0.98–0.99 | **0.001** |
| **Visit type** 1st | Ref | | | | | |
| 2nd | 1.83e-06 | 0 | 0.977 | 4.64E-07 | 0 | 0.988 |
| 3rd | 0.77 | 0.45–1.31 | 0.342 | 0.69 | 0.38–1.24 | 0.218 |
| 4th | 1.27 | 0.21–7.61 | 0.793 | 0.51 | 0.05–5.02 | 0.57 |
| **Entry point** VIA routine check-up | Ref | | | | | |
| ART clinic | 0.94 | 0.80–1.11 | 0.510 | 0.88 | 0.72–1.09 | 0.27 |
| Other | 0.62 | 0.40–0.95 | **0.031** | 0.68 | 0.39–1.17 | 0.169 |
| **Sample type** Self-collected | Ref | | | | | |
| Provider collected | 2.15 | 1.69–2.74 | **<0.001** | 2.15 | 1.66–2.79 | **<0.001** |
| **HIV status** Negative | Ref | | | | | |
| Positive | 1.65 | 1.24–2.18 | **<0.001** | 1.77 | 1.22–2.55 | **0.002** |

smaller sample size (145) and only included WLWHIV. Therefore, the inclusion of non-uniform hr-HPV types in the two previous studies could in part explain the differences in the findings. The comparability of our study and the two previous studies may also have been affected by time differences and changes in disease trends especially since both were done more than 10 years ago [22–25].

We observed WLWHIV that HIV infection was significantly associated with HPV infection. These findings are consistent with previous studies in the region, emphasizing the vulnerability of WLWHIV to HPV infection [26,27]. HIV infection is thought to disrupt epithelial tight junctions which may promote both the establishment of new HPV infection and progression of existing infection in the female genital tract [28]. While HPV infection naturally clears in most of the infected women, HPV infection commonly persists in the WLWHIV due to impaired immune responses resulting from HIV pathology [29]. The higher prevalence of HPV infection in this population highlights the urgent need for targeted interventions, such as HPV vaccination and consistent periodic cervical cancer screening, to reduce the burden of HPV-related diseases among WLWHIV in Zambia.

Another significant finding of our study was the association between age and HPV infection. Both univariate and multivariate analyses showed a consistent trend, indicating that increasing age was associated with a reduced likelihood of having HPV infection. This finding aligns with previous studies conducted in SSA, and globally, and underscores the importance of age as a significant factor in HPV epidemiology [30–33]. A study in Zambia found that young women were less likely to use condoms in sexual intercourse compared to their male counterparts [34]. Another study in South Africa showed that older women are more likely to use a condom during sexual intercourse compared to young women [35]. Condom use has been shown to offer protection against HPV infection, thus non-condom use during sexual intercourse exposes young women to HPV women, which may in part explain the higher prevalence of HPV in young women [36,37]. Additionally, young women are more likely to have multiple sexual partners compared to older women, which puts them at higher risk of contracting HPV infection [38–41]. Understanding the age-specific patterns of HPV infection can inform age-targeted preventive strategies, including vaccination programs and screening guidelines [42,43].

Another noteworthy finding in our study was the influence of the sample collection method on HPV infection rates. Women who had their samples collected by a healthcare provider had a significantly higher prevalence of HPV infection compared to the self-collected ones. This could be because health care providers are trained and are more likely to get an accurate sample whereas patients on the other hand may not fully understand proper sample collection. These findings are not consistent with other studies that have shown the reliability of self-collected HPV samples [44–47]. However, on the other hand, the findings agree with a few studies that found that the Hologic Panther Aptima kit-based HPV tests had less sensitivity on self-collected samples [48,49]. While these findings cannot be used to determine the actual reliability of self-collected HPV samples, our findings suggest that the sample collection technique may impact the detection of HPV infection in the Zambian population. These findings stress the need to study the effectiveness of new diagnostic methods in different populations and settings before rolling out the methods. Furthermore, the findings have practical implications for healthcare providers and emphasize the importance of standardized protocols and training for sample collection to ensure accurate and reliable results.

## Strengths and limitations

Our study has several strengths, including a large sample size and the inclusion of WLWH, which enhances our understanding of HPV infection in a high-risk population. To our knowledge, our study is the first study with a large sample size to investigate the prevalence of HPV in Zambia. Large sample sizes are advantageous in that they are more representative of the population when compared with studies with smaller numbers.

Majority of the participants were WLWH, indicating the importance of studying HPV infection in this particular group. It is well-established that individuals with compromised immune systems, such as WLWH, are at a higher risk of acquiring HPV infection and developing associated complications, including cervical cancer [1,50–52]. By including a substantial number of WLWHIV in our study, we have shed light on an important aspect of HPV epidemiology within the Zambian context.

Despite the strengths, we acknowledge that our study had several limitations. Firstly, since our research utilized data provided on laboratory request forms, the study was limited to the variables provided by the laboratory forms, thus we could not include variables that would have provided a more comprehensive understanding of the factors associated with HPV infection such as the age at first coital experience, sexual partners, smoking, alcohol consumption, and other underlying conditions. Secondly, we could also not perform HPV genotyping due to restricted resources, thus we could not determine the most prevalent hr-HPV types. Thirdly, our study was conducted solely within cancer clinics in a single province, which may have restricted the generalizability of our findings to the broader population of Zambia. Finally, the cross-sectional nature of our study design precludes us from establishing causal relationships between the identified factors and HPV infection.

## Conclusion

In conclusion, our study contributes to the existing body of knowledge on HPV infection in SSA, particularly in the context of Zambia. The higher prevalence of HPV infection among WLWHIV and the association between age and infection highlight the need for targeted interventions and age-specific preventive strategies. The impact of sample collection methods on HPV detection emphasizes the importance of standardized sample collection protocols for both clients and healthcare workers. Longitudinal studies and a broader representation of the

population are warranted to enhance our understanding of HPV epidemiology and inform effective preventive measures.

## Supporting information

**S1 File.**
(PDF)

**S1 Data. Prevalence and correlates of human papillomavirus infection in females from Southern Province, Zambia study.**
(XLSX)

## Acknowledgments

We express our heartfelt gratitude to the management of Livingstone University Teaching Hospital and the dedicated laboratory staff for their invaluable support during the data collection process. Their unwavering assistance, professionalism, and commitment greatly facilitated our research, enabling us to complete this study effectively. We appreciate their cooperation and the resources provided, which were essential to the success of our work.

## Author Contributions

**Conceptualization:** Lweendo Muchaili, Precious Simushi, Sepiso K. Masenga.

**Data curation:** Lweendo Muchaili, Precious Simushi, Bislom C. Mweene, Tuku Mwakyoma, Sepiso K. Masenga, Benson M. Hamooya.

**Formal analysis:** Lweendo Muchaili, Sepiso K. Masenga, Benson M. Hamooya.

**Investigation:** Lweendo Muchaili, Precious Simushi, Bislom C. Mweene, Tuku Mwakyoma, Sepiso K. Masenga, Benson M. Hamooya.

**Methodology:** Lweendo Muchaili.

**Project administration:** Lweendo Muchaili.

**Supervision:** Precious Simushi, Sepiso K. Masenga, Benson M. Hamooya.

**Writing – original draft:** Lweendo Muchaili, Precious Simushi, Bislom C. Mweene, Tuku Mwakyoma.

**Writing – review & editing:** Lweendo Muchaili, Precious Simushi, Bislom C. Mweene, Tuku Mwakyoma, Sepiso K. Masenga, Benson M. Hamooya.

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
