## [Decision Letter · Decision Letter 0]

15 Apr 2024

PONE-D-24-06185Prevalence and correlates of Human Papillomavirus infection in females from Southern Province, Zambia: A cross-sectional studyPLOS ONE

Dear Dr. Muchaili,

Thank you for submitting your manuscript to PLOS ONE. After careful consideration, we feel that it has merit but does not fully meet PLOS ONE’s publication criteria as it currently stands. Therefore, we invite you to submit a revised version of the manuscript that addresses the points raised during the review process.

We look forward to receiving your revised manuscript.

Kind regards,

Kehinde S. Okunade

Academic Editor

PLOS ONE

Journal Requirements:

2. We note that your Data Availability Statement is currently as follows: All relevant data are within the manuscript and its Supporting Information files

Reviewers' comments:

Reviewer's Responses to Questions

**Comments to the Author**

1. Is the manuscript technically sound, and do the data support the conclusions?

Reviewer #1: Yes

Reviewer #2: Yes

2. Has the statistical analysis been performed appropriately and rigorously? 

Reviewer #1: Yes

Reviewer #2: Yes

3. Have the authors made all data underlying the findings in their manuscript fully available?

Reviewer #1: Yes

Reviewer #2: Yes

4. Is the manuscript presented in an intelligible fashion and written in standard English?

Reviewer #1: Yes

Reviewer #2: Yes

5. Review Comments to the Author

Reviewer #1: Thank you for asking me to review the article titled “Prevalence and correlates of Human Papillomavirus infection in females from Southern Province, Zambia: A cross-sectional study”.

The study was a cross sectional retrospective study conducted in Zambia. It involved collection of data on 4, 612 included participants that had been tested for HPV infection between September 2021 and August 2022 using the Aptima® HPV kit on the Hologic Panther platform. The aim was to determine the prevalence and correlates of HPV infection among women from Southern province using programmatic data to generate hypotheses for follow-up interventional studies. The primary outcome was 35.56% of the participants had HPV infection. Significantly age, HIV status and method of sample collection were important correlates for HPV infection among the studied population.

I must commend the team on the research work, as the area of study is of clinical significance to women’s health. Their study contributes to the knowledge of the overall prevention of cervical cancer and has been properly conducted.

The study methodology was sound with ease of reproducibility. The data analysis was comprehensive, and the study design was appropriate. I noted minor corrections, but these do not affect the acceptance of the article.

1. Appropriate punctuation marks are required as some have been omitted.

2. They chose to use the Wilcoxon rank sum text in place of the initially stated Shapiro-Wilk test, this should be clarified.

3. The authors should review the entire manuscript for grammar, typo and punctions.

Overall, the study presents results of an original research with a retrospective cross-sectional design on the Prevalence and correlates of Human Papillomavirus infection in females from Southern Province, Zambia. The methods have been documented properly with ease of reproducibility and with sufficient detail. The results have been represented appropriately and the conclusion was written to support results from the data. The article was well written with good scientific flow. The research integrity was sufficient. I recommend the manuscript be accepted. Thank you.

Reviewer #2: The authors should review and provide a detailed description of the results presented in Table 1. The table currently does not include all participants and lacks an explanation for this omission. It would be helpful if the authors could include a guideline or statement/checklist to ensure transparency and provide a comprehensive description of the entire study.

6. PLOS authors have the option to publish the peer review history of their article (what does this mean?). If published, this will include your full peer review and any attached files.

Reviewer #1: No

Reviewer #2: No

---

## [Author Response · Author response to Decision Letter 0]

4 Jun 2024

Mulungushi University,

P.O Box 60009,

Livingstone.

23rd April 2024.

The Reviewers,

PLOS ONE Journal.

Dear PLOS ONE reviewers,

RE: RESPONSE TO REVIEWER COMMENTS

Thank you so much for reviewing our manuscript and providing invaluable reviews. We appreciate your review comments which we believe have improved our manuscript's readability and credibility.

We have taken time to correct our manuscript and provided a manuscript copy with track changes to reflect our work on the manuscript.

Below are our responses to the review comments.

Response from authors: Thank you. We have referred to the comments and have since adjusted the naming of the manuscript and supporting documents to PLOS ONE’s style requirements.

2. We note that your Data Availability Statement is currently as follows: All relevant data are within the manuscript and its Supporting Information files

Response from authors: Thank you, we have since attached the raw data to this submission.

Response from authors: Thank you, we have since deleted the ethics statement in other parts of the manuscript and only retained it in the methods section.

Response from authors: Thank you, we have since made the captions as per recommendation.

Response from authors: Thank you, we have reviewed our reference list as recommended. We have not cited any retracted article.

5. Review Comments to the Author

Reviewer #1: Thank you for asking me to review the article titled “Prevalence and correlates of Human Papillomavirus infection in females from Southern Province, Zambia: A cross-sectional study”.

The study was a cross sectional retrospective study conducted in Zambia. It involved collection of data on 4, 612 included participants that had been tested for HPV infection between September 2021 and August 2022 using the Aptima® HPV kit on the Hologic Panther platform. The aim was to determine the prevalence and correlates of HPV infection among women from Southern province using programmatic data to generate hypotheses for follow-up interventional studies. The primary outcome was 35.56% of the participants had HPV infection. Significantly age, HIV status and method of sample collection were important correlates for HPV infection among the studied population.

I must commend the team on the research work, as the area of study is of clinical significance to women’s health. Their study contributes to the knowledge of the overall prevention of cervical cancer and has been properly conducted.

The study methodology was sound with ease of reproducibility. The data analysis was comprehensive, and the study design was appropriate. I noted minor corrections, but these do not affect the acceptance of the article.

1. Appropriate punctuation marks are required as some have been omitted.

Response from authors: Thank you so much. We have since worked on the punctuation marks as recommended.

2. They chose to use the Wilcoxon rank sum text in place of the initially stated Shapiro-Wilk test, this should be clarified.

Response from authors: Thank you. We apologize for the omission, we should have indicated that we used the Shapiro-Wilk test to test for normality, and used Wilcoxon rank sum test to test for the relationship between the medians of the two outcome groups. We have since included this statement.

3. The authors should review the entire manuscript for grammar, typo and punctions.

Overall, the study presents results of an original research with a retrospective cross-sectional design on the Prevalence and correlates of Human Papillomavirus infection in females from Southern Province, Zambia. The methods have been documented properly with ease of reproducibility and with sufficient detail. The results have been represented appropriately and the conclusion was written to support results from the data. The article was well written with good scientific flow. The research integrity was sufficient. I recommend the manuscript be accepted. Thank you.

Response from authors: Thank you so much. We have since worked on the typos and punctuation marks as recommended. We have attached the manuscript with the tracked changes for your review.

Reviewer #2: The authors should review and provide a detailed description of the results presented in Table 1. The table currently does not include all participants and lacks an explanation for this omission. It would be helpful if the authors could include a guideline or statement/checklist to ensure transparency and provide a comprehensive description of the entire study.

Response from authors: Thank you, we have since reviewed Table 1, and further provided a detailed description accounting for the missing variable entries. Note that we have also added a column (n) to account for the missing variable entries in Table 1.

Yours,

Lweendo Muchaili

On behalf of the authors

---

## [Editor Report · Decision Letter 1]

21 Jun 2024

Prevalence and correlates of Human Papillomavirus infection in females from Southern Province, Zambia: A cross-sectional study

PONE-D-24-06185R1

Dear Dr. Muchaili,

We’re pleased to inform you that your manuscript has been judged scientifically suitable for publication and will be formally accepted for publication once it meets all outstanding technical requirements.

Kind regards,

Kehinde S. Okunade

Academic Editor

PLOS ONE

Additional Editor Comments (optional):

Accept manuscript
---

## [Editor Report · Acceptance letter]

27 Jun 2024

PONE-D-24-06185R1 

PLOS ONE

Dear Dr. Muchaili, 

I'm pleased to inform you that your manuscript has been deemed suitable for publication in PLOS ONE. Congratulations! Your manuscript is now being handed over to our production team.

Kind regards, 

on behalf of

Dr. Kehinde S. Okunade 

Academic Editor

PLOS ONE